# Effects of Continuous and Pulsed Ultrasonic Treatment on Microstructure and Microhardness in Different Vertical Depth of ZL205A Castings

**DOI:** 10.3390/ma13194240

**Published:** 2020-09-23

**Authors:** Gang Lu, Yisi Chen, Qingsong Yan, Pengpeng Huang, HongXing Zhan, Yongbiao Duan

**Affiliations:** National Key Laboratory of Light Alloy Processing Science and Technology, Nanchang Hangkong University, Nanchang 330063, China; lugang@nchu.edu.cn (G.L.); 2003080503105@stu.nchu.edu.cn (Y.C.); 1703080503009@stu.nchu.edu.cn (P.H.); 1803085201030@stu.nchu.edu.cn (H.Z.); 150308050315@stu.nchu.edu.cn (Y.D.)

**Keywords:** continuous ultrasonic treatment, pulsed ultrasonic treatment, microstructure, grain refinement, segregation, ZL205A alloy

## Abstract

In this paper, in order to improve the performance of the ZL205A castings, continuous ultrasonic and pulsed ultrasonic treatments were applied to the melted alloy to study the effect of ultrasound propagation distance on microstructure and microhardness. The results indicated that ZL205A grains were significantly refined by ultrasonic vibration, but the refinement effect became weak gradually with the increase of sampling depth. The minimum grain sizes were 103.2 μm and 122.5 μm respectively in continuous and pulsed ultrasonic treatment. Grain boundary segregation also became more serious and coarser with the increase of vertical depth. In addition, microhardness and vertical depth are not positively correlated linearly. As the vertical depth increased, microhardness first decreased and then increased, the maximum hardnesses were 73.9 HV and 72.84 HV, respectively, in the two process modes. According to the experiment results and available studies, the mechanism of ultrasonic treatment maybe that: the cooling rate of solidification interface front increased by cavitation and streaming, thus changing the solute redistribution behavior of the ZL205A melt.

## 1. Introduction

ZL205A alloy is widely used in aviation manufacturing due to its advantages of high specific strength, high specific stiffness, and high toughness. However, it is difficult to consistently produce precision parts by traditional casting methods due to the existence of microporosity and coarse microstructures and microsegregation [1,2,3]. Microsegregation is the non-uniform distribution of solute atoms, which is an inevitable phenomenon in the solidification process of alloy [4]. Copper atoms in ZL205A tend to gather to the grain boundary of primary phase and form coarse skeletal segregation [5,6]. The mechanical behavior (in particular creep resistance) of aluminum alloys depended strongly on the distribution of the copper atoms. Segregation structure seriously reduced the uniformity of castings, and would not be improved after homogenizing annealing, even separating out a heterogeneous sediment lead to stress concentration and crack initiation. Those have a great impact on mechanical properties, corrosion resistance, physical properties, and castability of ZL205A alloy [7,8,9,10].

The redistribution of solute atoms during solidification is affected by melt flow. Effective stirring can be used to control the miscrostructure and microsegregation of the alloy, such as mechanical stirring, electromagnetic stirring, and ultrasonic vibration [11,12]. Ultrasonic treatment (UST) is a new casting technology with no pollution, high efficiency, and low energy consumption. Past studies have shown that ultrasonic waves can remarkably refine microstructure and improve mechanical performance because of cavitation and ultrasonic streaming [13]. Collapse of cavitation bubbles increased the local temperature and pressure and resulted in structural fluctuation and local supercooling. Streaming would disperse the aggregate in the melt and fuse dendrites to produce small particles, which can act as the core of heterogeneous nucleation. Under the synergistic effect of cavitation and streaming, the nucleation rate of alloy was improved during cooling and solidification process [14]. After intense streaming, the segregation would be destroyed and reducing the grain boundary segregation and the microhardness of samples would also change because of ultrasonic treatment [15,16]. Our previous research has indicated that pulsed UST can change the solute redistribution behavior of ZL205A alloy melt [4].

The effect of UST on the microstructure of aluminum alloy has been extensively studied abroad. However, there are few reports about the influence of ultrasound propagation distance on aluminum alloy melt, and the mechanism of UST is also rarely discussed [17,18,19,20,21]. In this paper, the effects of continuous and pulsed ultrasonic treatments on the solidification structure of ZL205A alloy were studied, respectively. The differences of two ultrasonic processes were simply compared, the microstructure and segregation at different ultrasonic power and vertical depth were investigated by scanning electron microscope (SEM) and energy disperse spectroscopy (EDS). From the perspective of diffusion and solute redistribution, analyzing the relationship of solidification structure as well as microhardness and vertical sampling depth could provide theoretical guidance for improving the ultrasonic melt treatment process in casting.

## 2. Materials and Methods

### 2.1. Experimental Material

The experimental material was the ZL205A aluminum alloy with the chemical composition listed in Table 1.

### 2.2. Experimental Equipment and Procedures

The ultrasonic system was adjusted by the fixing bracket to ensure the vibration tool head was in the central position of the coated sand casting mold and the ultrasonic vibration tool head was immersed in 30 mm of ZL205A alloy melt as shown in Figure 1. The ultrasonic vibration system includes an ultrasonic transducer, amplitude transformer, and vibration tool head. Both ultrasonic vibration devices have a frequency of 20.1 KHz and a power range of 600–2000 W. Specific process parameters of continuous ultrasonic and pulsed ultrasonic are shown in Table 2 and Table 3, respectively. Ultrasonic vibration was applied before the surface of liquid metal has been solidified. Defined the longitudinal direction from top to bottom as the vertical depth, and the ultrasonic vibration tool head was taken as the starting point.

The coated sand mold used for casting was preheated to 80–120 °C. The ZL205A aluminum alloy ingot was melted in a graphite crucible inside an electric resistance furnace. The melt was smelted and refined for five minutes with 0.8 wt % refining agent at 730 °C and then raised the temperature of molten metal to 760 °C. Transferred the liquid metal to the coated sand mold after removing scum. Start the ultrasonic vibration device when the alloy melt temperature drops to 730 °C. A total of five samples were taken and each spaced 50 mm apart after the casting was completely cooled.

### 2.3. Test Method

The microstructure characterization samples were polished by grinding paper from 800 to 2000 grit and metallographically polished with 1 μm alumina, and subsequently etched using the Keller’s reagent (90 mL H_2_O, 2.5 mL HNO_3_, 1.5 mL HCl, 1.0 mL HF). Metallographic observation were conducted with a XJP-6A optical microscope (OM, Jinan Zhongte testing machine Co.LTD, Jinan, China). Primary phase and segregation were observed with the use of a scanning electron microscope (SEM, Quanta 200, FEI, Czech Republic). Energy disperse spectroscopy (EDS, Oxford INCA, UK) was used to characterized the distribution and content of aluminum and cooper. The micro-vickers hardness of the sample was tested by Vickers(DureScan, Denmark). Five different grains were selected for each sample to test the microhardness, and the results were average several times.

## 3. Results and Discussion

### 3.1. Microstructure of Different Vertical Depth Under Continuous and Pulsed Ultrasonic Treatment

The microstructure characteristics at different vertical depths and the results are respectively shown in Figure 2. It can be seen that the solidification structure of ZL205A alloys changed obviously with the increase of vertical depth. The grain size of samples gradually become larger both in continuous and pulsed UST at a power of 1200 W. However, at the same vertical depth of 100 mm and 150 mm, the refinement of pulsed UST is better. For continuous UST, the smallest grain size is at a depth of 0 mm, and the maximum grain size is 153.7 μm at a depth of 200 mm. For pulsed ultrasonic processing, the grain size at the depths of 0 mm and 200 mm are 126.5 μm and 152.3 μm respectively. However, the results show that the refining effect of UST on solidification structure of ZL205A alloy is not obvious at a deep depth in Figure 2e,j.

When ultrasonic treatment was applied to ZL205A alloy melt, part of the ultrasonic energy was converted into heat energy due to the friction between different particles because of the viscous force of melt media. Heat conduction in melt medium made heat exchange between dense and sparse areas of melt, resulting in loss of sound energy. On the macro level, the ultrasonic attenuation reduced with the extension of propagation distance. The attenuation modes mainly include scattering attenuation, diffusion attenuation, and absorption attenuation. That attenuation law is shown in Formula (1) [22]
(1)$$P=P_{0}e^{-\alpha x}$$
in which *P*_0_ and *P* are the initial ultrasonic pressure and ultrasonic pressure at a propagation distance of *x*, respectively. Where α is attenuation coefficient, and *x* is propagation distance. As we can see from Formula (1), when x increases to a extremely large, *P* would decrease and close to zero, which means ultrasonic have a few influences on this area. As we can see in Figure 2, the differences of micromorphological between depth of 150 mm and depth of 200 mm were very small. However, the thinning effect of pulsed UST was more significant than that of continuous UST at the depth of 100 mm and 150 mm. This phenomenon indicates that the attenuation degree of pulsed ultrasonic treatment was lower than that of continuous ultrasonic treatment during the propagation process in ZL205A alloy melt.

### 3.2. Grian Size at Different Vertical Depth under Continuous and Pulsed Ultrasonic Treatment

After applied ultrasonic vibration, grains in each part of the longitudinal region were refined obviously as shown in Figure 3 and Figure 4. As opposed to experimental group, the grain size became smaller at a deeper place without UST, because liquid metal of the sample at the bottom of casting mould would first touch coated sand during pouring process and then crystallize and solidify preferentially under the action of chilling. At the same depth, the refinement improved with the higher vibration power. At the same power, the grain size of ZL205A alloy presented an increasing trend with the increase of vertical depth.

Microstructural modification and refinement by UST can be clearly observed. A plausible explanation is that high pressure made the equilibrium solidification temperature decreased by cavitation. Local high temperature and high pressure lead to the increase of effective supercooling in cavitation area. Accordingly, heterogeneous nucleation was promoted by UST during solidification. In addition, the fine free grains in alloy melt were dispersed to other parts of the casting due to the ultrasonic streaming and served as heterogeneous nucleation cores.

For continuous UST, the minimum grain size was 103.02 μm at the vertical depth of 0 mm and at a power of 1500 W. In the same position, the grain size was 113.24 μm at the maximum power of 1800 W. The possible reason for this phenomenon may be that continuously increase of ultrasonic power lead to the alloy melt overheating, resulting a decrease of the supercooling. At the same time, the solidification time of ZL205A melt will be prolonged and most of gains grew slowly. When the effects of heat were greater than that of cavitation and streaming the solidification structure would be coarsened [23]. For pulsed ultrasonic treatment, the minimum grain size was 122.54 μm at a power of 1800 W and the depth of 0 mm. The grain size and vertical depth are positively correlated linearly at each power. Furthermore, the grain size of the sample at the bottom was less refined and was not well refined at a low vibration power. Therefore, to refine the solidification of the structure in whole casting, a higher ultrasonic vibration power needs to be applied. However, excessive power would lead to local overheating and solidification defection defects make grain coarsened again, so that appropriate technological parameters of UST should be selected comprehensively. 

Figure 5 shows a comparison of measured grain size in the continuous and pulsed ultrasonic treatments. It can be seen from this chart that the grain size of samples with pulsed UST are smaller than that of continuous UST at the same longitudinal depth. As can we can see, the two curves were similar, and there is no essential difference between the two treatments in mechanism. Difference in refining effect of grain size was due to the effective action strength. When the average output power of the ultrasonic transmitting power is consistent, the peak power and sound pressure of pulsed ultrasonic treatment is higher than that of continuous ultrasonic treatment. That is to say, the pulsed ultrasonic intensity is higher than the continuous ultrasonic intensity at the same area.

### 3.3. Segregation at Different Vertical Depth Under Pulsed Ultrasonic Treatment

The microstructure of ZL205A aluminum alloy by 1200 W pulsed UST was observed by SEM, and the results are shown in Figure 6.

As shown in Figure 6, the dark gray tissue is α-Al matrix, and the light gray tissue at grain boundaries are cooper segregation. The segregation in Figure 6a are scattered fragments, and look skeletal in Figure 6b. The segregation is in the shape of ring stents in Figure 6c at the depth of 200 mm. The segregation tissue at the grain boundary becomes larger with the increase of vertical depth. During the cooling process of ZL205A alloy melt, some aluminum atoms would agglomerate and crystallize, forming a primary α-Al phase. With the formation of α-Al phase, the content of copper in the remaining melt raised. Eutectic reaction occurred and eutectic Al_2_Cu was formed at 548 °C when the concentration reaches the eutectic point [24,25]. At the same time, copper and aluminum atoms in the eutectic phase were separated at the grain boundary and mutual dislocation, then the separated copper atoms concentrated at the boundary of α-Al matrix, forming grain boundary segregation at last. The mass fraction of copper in light gray segregation structure was far greater than that of ZL205A matrix. At the end of crystallization, a skeleton-shaped segregation structure was formed in the grain boundary.

The EDS analysis results of spectrum 1 and spectrum 2 in Figure 6 are shown in Figure 7, respectively, and spectrum 3 are shown in Figure 6. The mass fraction of copper and aluminum in boundary segregation is shown in Figure 8. According to the obvious results, segregation at the grain boundary was mainly composed of aluminum and copper, and the content of copper in the segregation increased with the increase of vertical depth. The matrix phase is composed entirely of aluminum. Most of the copper atoms in the ZL205A alloy were enriched at the grain boundary. The mass fraction of copper in the sample which was closest to the ultrasonic vibration tool head was about 52.31% and increased to 53.24% at the vertical depth of 100 mm, for the deepest area, the content of copper was 58.31%

The effect of ultrasonic treatments on element distribution can also be due to the cavitation and streaming. Ultrasound would produce powerful shock waves and heat flow in alloy melt because of a large number of cavitation bubbles collapsing. The primary segregation was broken up and remelted by the agitation of streaming. UST reduced the amount of segregation and improved the solid solubility and uniformity of solute elements in the grain interior.

Cavitation and streaming were most intense in the solidification interface front. Therefore, in middle and late stage of solidification, liquid phase between the grain boundaries is under a great pressure. After intense steaming, the first solidified segregation structure was destroyed, so that grain boundary segregation had changed from skeleton thickness to the fine fragment structure after being completely frozen. The redistribution low of copper in melt can be seen in Formula (2) [26]
(2)$$K_{E}^{}=\frac{k_{0}}{k_{0}+(1-k_{0})e^{\frac{-R}{D_{L}}\delta }}$$
in which *k_E_* and *k*_0_ are the effective distribution coefficient and equilibrium distribution coefficient, respectively. Where *R* is the solidification of solid phase rate, *δ* is the thickness of diffusion boundary layer, and *D_L_* is diffusion coefficient. We can know from Formula (2), there are three main factors affecting the solute distribution. Convection and diffusion in ZL205A would be enhanced by UST. The homogeneous mixing zone was expanded in the liquid phase because of the increased diffusion of the solute elements, and then compacted the solute enrichment layer. Which means *δ/D_L_* will be reduced made solute redistribution decrease. However, there was less copper segregation closer to the vibration tool head. The solidification rate was the most important influence factor about solute distribution, because experiment results showed that *K_E_* would be increased by more incentive UST. The mixing effect of streaming changes the distribution situation of flow field and equates the temperature field in ZL205A melt. As a result, the solute enrichment layer at the solidification interface front had a faster cooling rate so copper atoms had no time to be discharged from the solid phase [27]. Therefore, *k_E_* will be reduced with the vertical depth increased in our experiment and the closer to the bottom, the more copper segregation would be formed. In actual production, the selection of vibration location should be considered in many aspects to reduce segregation.

### 3.4. Microhardness at Different Vertical Depths Under Continuous and Pulsed Ultrasonic Treatment

The effects of different treatment processes and sampling locations on microhardness of ZL205A aluminum alloy is shown in Figure 9 and Figure 10. After applying ultrasonic vibration, the microhardness in each part of the longitudinal region have taken place a marked change. As we can see, microhardness increased with the increase of vertical depth in samples without ultrasonic treatment. Similar to the grain refinement mechanism, this phenomenon is due to the bottom melt’s first contact with the coated sand during pouring process. The finer microstructure and the denser tissue would be obtained by chilling and gravity, respectively, and both of them are beneficial for improving the microhardness.

Samples without ultrasonic vibration would have a better solid-solution strengthening effect because the uneven distribution of elements at the solid–liquid interface. Non-equilibrium solidification made degree of supersaturation of the matrix increasing during the solidification process. Grains of ZL205A alloy were obviously refined, and forming equiaxed grains after UST. Previous studies have shown that equiaxed crystals have higher microhardness than dendritic crystals [28]. Interconnections of dendrite produced shrinkage cavity and porosity, and there were more contact surfaces at grain boundaries. The interface with complex morphology was easy to cause defects. It can be seen from analyses above that the effect of ultrasonic waves on melt will increase the solidification rate of the front edge at solid–liquid interface. And a higher solidification rate can reduce the condition of constitutional supercooling increasing the range of constitutional supercooling and made the morphology of grains changed from dendritic crystal to round equiaxed crystal [26]. After UST, the number of grain boundaries per unit volume increased and there was more eutectic Al_2_Cu distributed around the grain boundary. The content of alloy elements inside grains was relatively decreased, so the solid-solution strengthening was weakened. Theoretically speaking, grain refinement helps to increase the hardness of the solidified structure, but at the same time increases the number of grain boundaries and results in solution strengthening of inside grains decreasing. Therefore, the influences of these factors on hardness were mutual restriction during UST.

For continuous ultrasonic treatment, the maximum hardness was 74 HV at the vertical depth of 0 mm at a power of 900 W. In the same region, it also achieved maximum hardness 72.84 HV in pulsed UST. With the increase of vertical depth, the microhardness curves first decreased and then increased, showing a ‘V’ shape. The lowest value of microhardness is usually obtained where the depth is 100 mm or 150 mm with UST. Under normal solidification, castings solidify from bottom to top, and got the highest density and the fewest defects at the depth of 200 mm. The temperature of vibration tool head was lower than alloy liquid even if it had been preheated and equivalent to a ‘chiller’ which would cool the surrounding melt. In addition, the ultrasonic streaming changed the temperature field in ZL205A alloy melt and made the center parts become the final solidification area.

Microhardness was decreased at the ultrasonic power of 600 W, the effect of UST was relatively weaker because cavitation was not strong enough. With the increase of ultrasonic power, the effect of UST was enhanced. Stronger streaming further promoted the convection of the whole melt and dissipated the latent heat of crystallization during the solidification stage through the coated sand wall and effectively homogenized the temperature field. With further increase of the ultrasonic power, grains of ZL205A alloy became smaller, so the solid solution strengthening of matrix structure will reduce according to the analysis above. At a power of 1800 W, ultrasonic treatment caused bottom melt overheating, and it became the final solidified zone. As we can see in Figure 8 and Figure 9, the microhardness was best in the ultrasonic power range of 900 W to 1200 W. As for 600 W and 1800 W, the micohardness was even lower than that of samples without UST.

## 4. Conclusions

In this paper, the microsturcture and microhardness in different vertical depths of ZL205A samples with UST were analyzed. The investigation led to the following conclusions:(1)Both continuous and pulsed ultrasonic treatment can refine the microstructure of ZL205A alloy effectively and, at the same vertical depth, refinement of pulsed ultrasonic treatment was better because of its higher effective sound intensity and less attenuation. Ultrasound energy is attenuated with the vertical depth increase and microstructure is coarser in the bottom.(2)Compared with samples without UST, the grain sizes decreased about 57.5% and 48.5% in the optimal refinement, respectively, under continuous and pulsed UST. The grain size became larger and segregation was much more serious with the increase of vertical depth.(3)The effective distribution coefficient (KE) of copper increased with the increased in cooling rate of solidification interface front under UST.(4)The optimum microhardness was 73.93 HV under continuous ultrasonic treatment, and it was 72.84 HV for pulsed ultrasonic treatment, compared with minimum microhardnesses increasing by 30.7% and 28.8% respectively. UST could change the crystal morphology from dendritic to equiaxed. With the increase of vertical depth, the microhardness curves first decreased and then increased, showing a ‘V’ shape.

## Figures and Tables

**Figure 1 materials-13-04240-f001:**
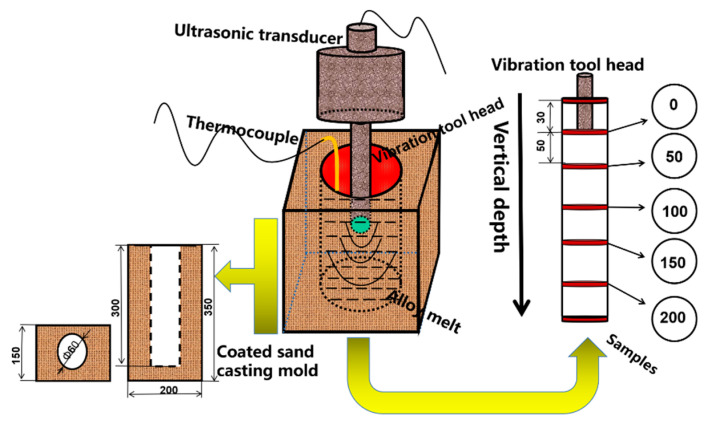
Experimental apparatus and sampling method.

**Figure 2 materials-13-04240-f002:**
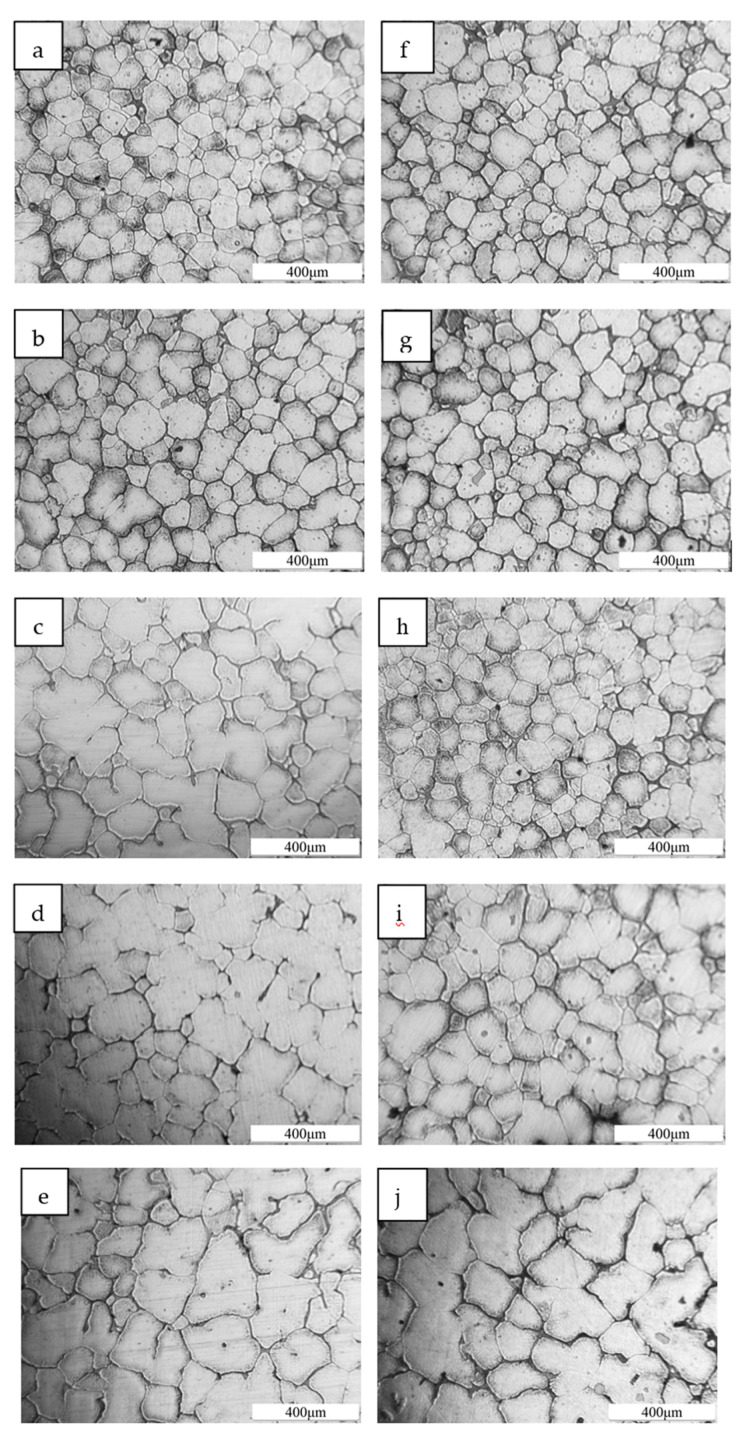
Solidification structure images of samples. Continuous ultrasonic 1200 W: (**a**) 0 mm, (**b**) 50 m, (**c**) 100 mm, (**d**) 150 mm, (**e**) 200 mm. Pulsed ultrasonic 1200 W: (**f**) 0 mm, (**g**) 50 m, (**h**) 100 mm, (**i**) 150 mm, (**j**) 200 mm.

**Figure 3 materials-13-04240-f003:**
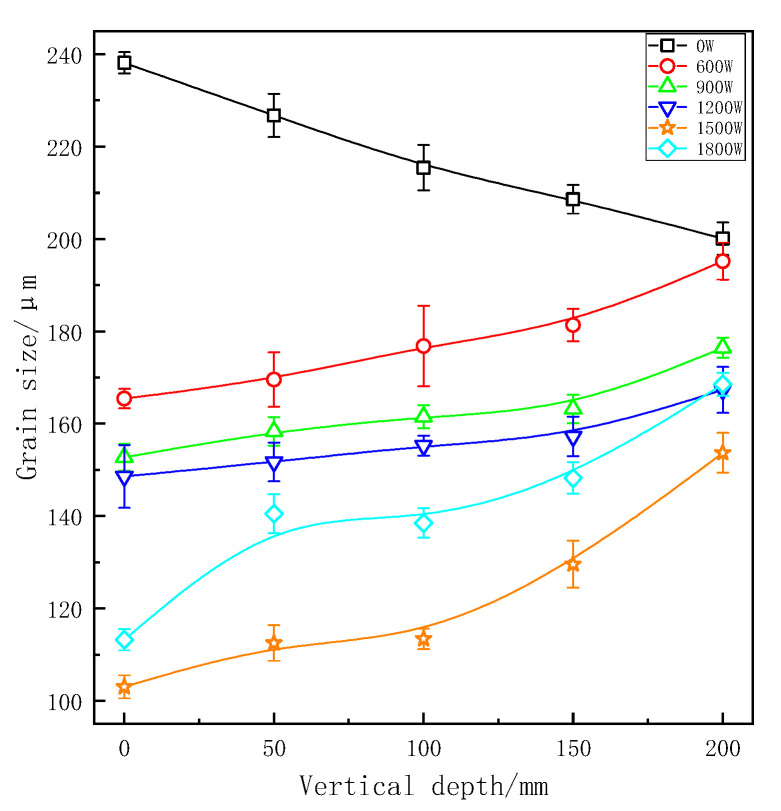
Grain size at different vertical depths under continuous UST.

**Figure 4 materials-13-04240-f004:**
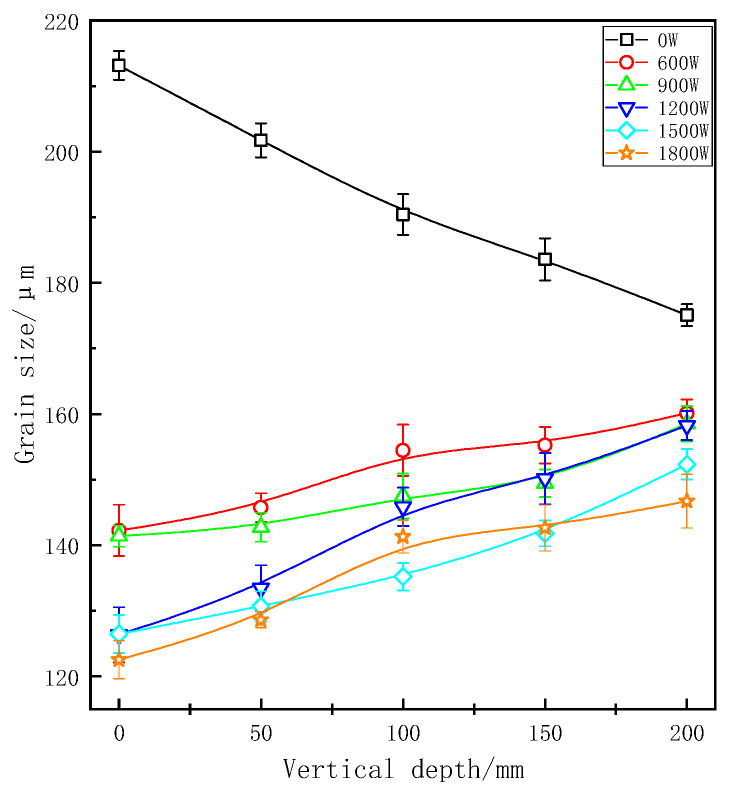
Grain size at different vertical depths under pulsed UST.

**Figure 5 materials-13-04240-f005:**
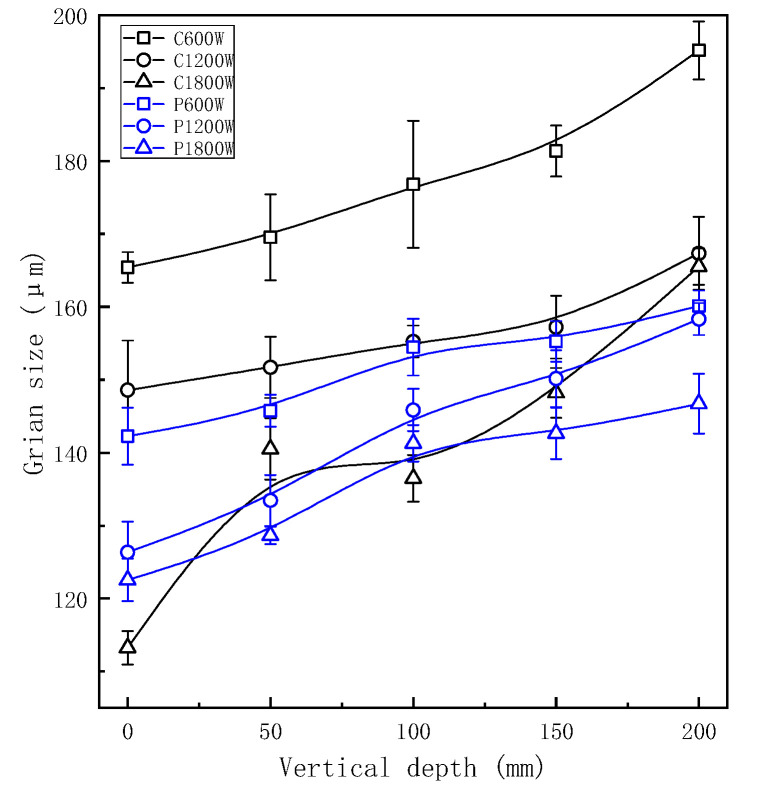
Comparison of grain size between the two processes.

**Figure 6 materials-13-04240-f006:**
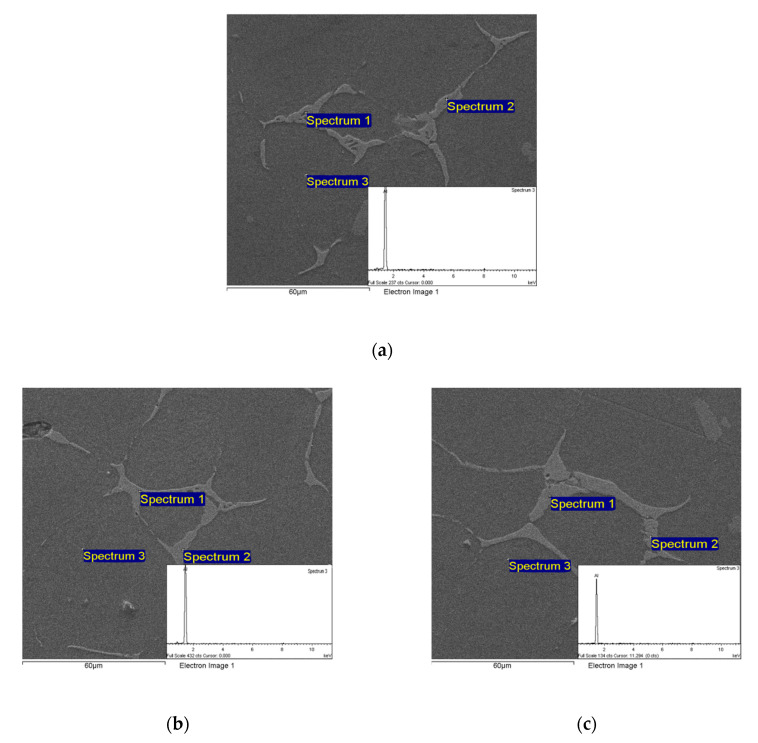
SEM image of ZL205A alloy; at depths of (**a**) 50 mm, (**b**) 100 mm, and (**c**) 200 mm.

**Figure 7 materials-13-04240-f007:**
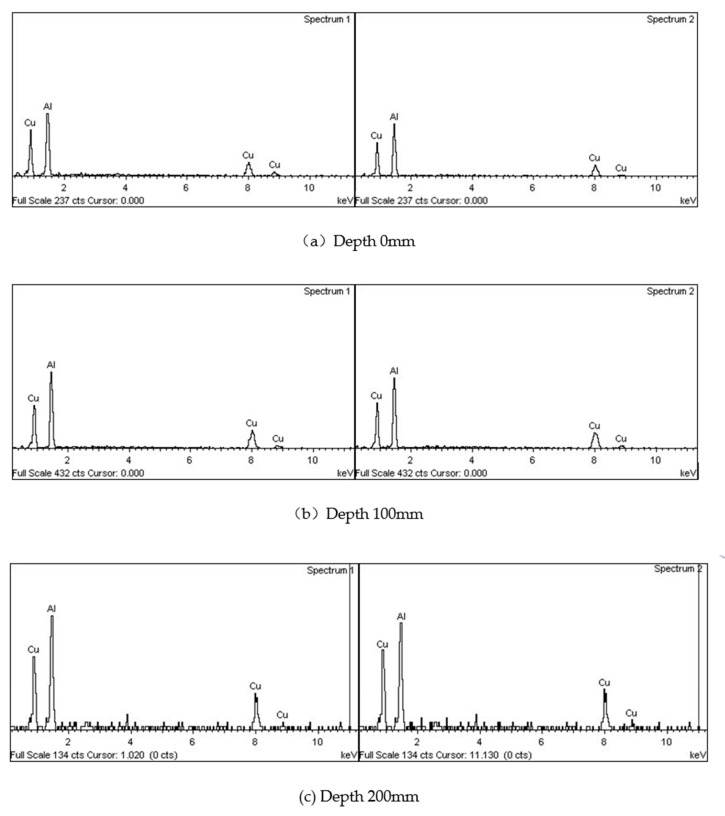
EDS analysis results of segregation at different vertical depths. (**a**) depth 0 mm; (**b**) depth 100 mm; (**c**) depth 200 mm.

**Figure 8 materials-13-04240-f008:**
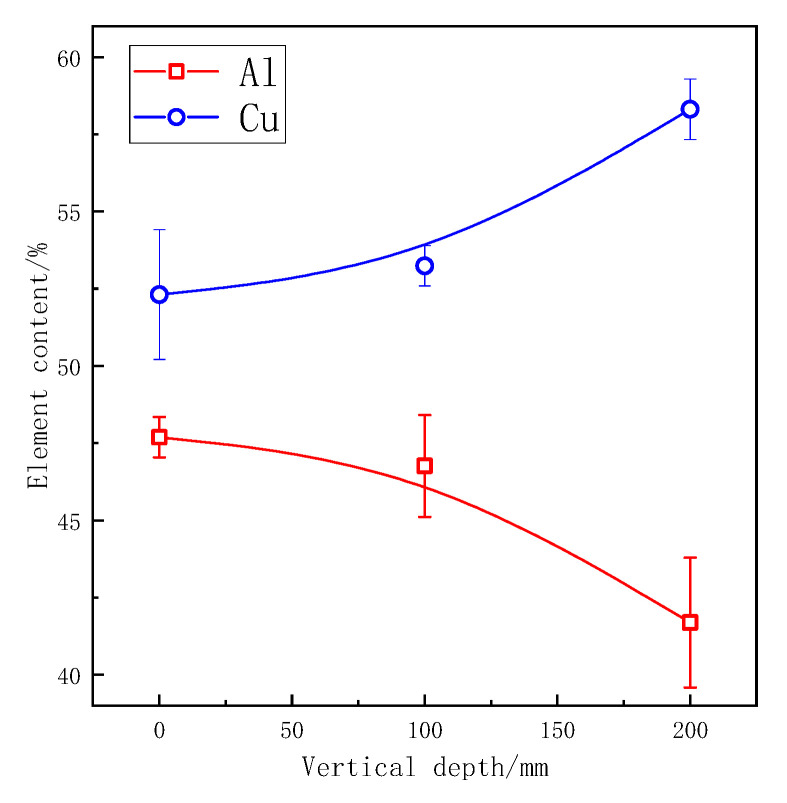
Element content of Al and Cu in segregation.

**Figure 9 materials-13-04240-f009:**
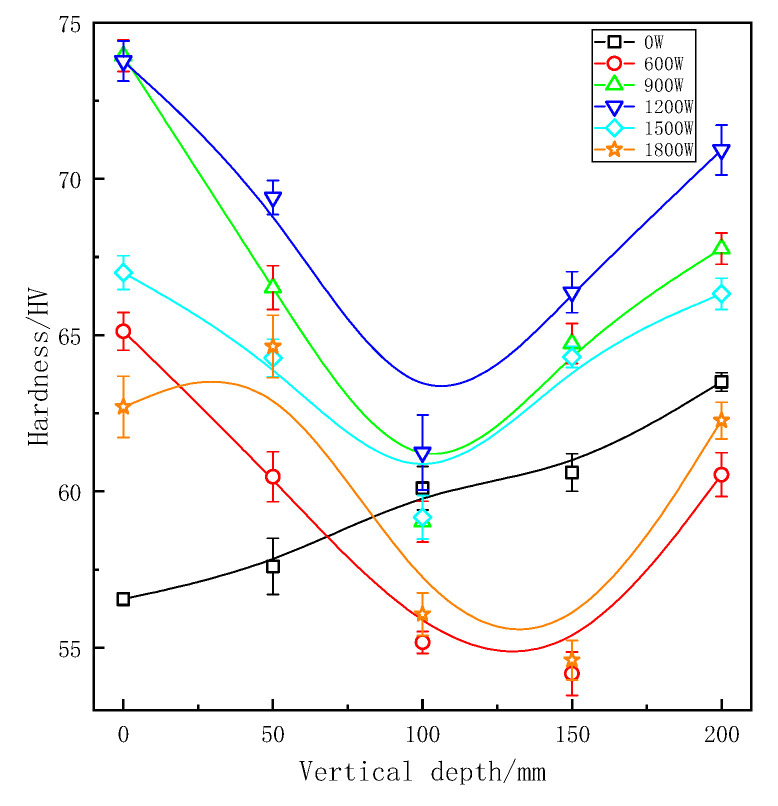
Microhardness at different depths under continuous ultrasonic treatment.

**Figure 10 materials-13-04240-f010:**
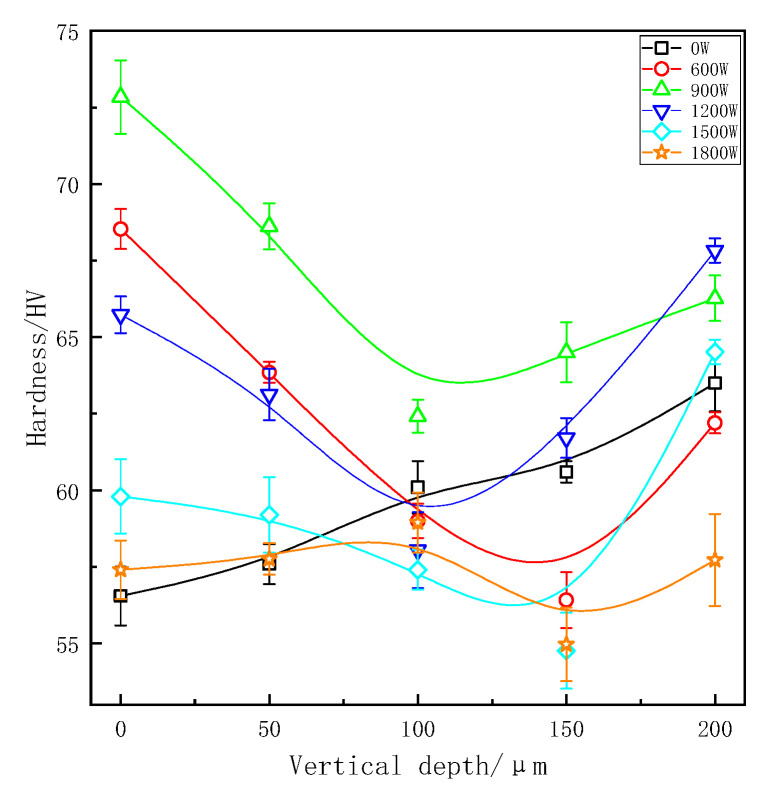
Microhardness at different depths under pulsed ultrasonic treatment.

**Table 1 materials-13-04240-t001:** Chemical composition of the ZL205A aluminum alloy (wt %)

Material	Cu	Mn	Ti	Cd	V	Zr	B	Al
ZL205A	4.6~5.3	0.3~0.5	0.15~0.35	0.15~0.25	0.05~0.3	0.05~0.2	0.005~0.06	Bal (remain part)

**Table 2 materials-13-04240-t002:** Process parameters of continuous UST.

Serial Number	Vibration Power (W)	Pouring Temperature (°C)	Vibration Frequency (KHz)
1	600	720	20.1
2	900		
3	1200		
4	1500		
5	1800		

**Table 3 materials-13-04240-t003:** Process parameters of pulsed UST

Serial Number	Vibration Power (W)	Pouring Temperature (°C)	Impulse Frequency (Hz)	Duty Cycle (%)	Vibration Frequency (KHz)
1	600	720	5	50	20.1
2	900				
3	1200				
4	1500				
5	1800				

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
