# Peer review of "Effects of Continuous and Pulsed Ultrasonic Treatment on Microstructure and Microhardness in Different Vertical Depth of ZL205A Castings"

_materials, 2020, doi:10.3390/ma13194240_

Round 1

Reviewer 1 Report

This paper seems fine. The experiment could maybe have done with some repeating to see what the intrinsic variation between runs is, compared to the effects of the insonification. Important: "Ultrasonic" is an adjective. Please replace in the title and through the paper with "Ultrasound".

Author Response

Dear reviewer:

   Thank you for your reply and suggestions and I have finished the modification. 

Reviewer 2 Report

In this manuscript, the authors examined how the ultrasonic treatment changes the structure and hardness of aluminum alloy. The structure of the alloy has been investigated by the microscopes and the hardness has been examined under various conditions of ultrasonic treatments. It seems that the most results have been consistently interpreted, however, I would like to comment several minor points, which I believe will improve the scientific soundness.

Line 125-126: In Figure 3 and 4, it seems that the grain size has relatively significant difference between the samples at 150 mm and at 200 mm. I suppose that the micromorphological feature at different depth should be discussed based on Figure 3 and 4, but not Figure 2. Even if the grain size does not linearly correlate with the ultrasonic pressure, it seems that the formula 1 is not enough to interpret the results shown in Figure 3, 4 (e.g. The grain size at different depth under continuous UST, especially at 1500 W, seems to increase at an accelerating rate.).

Figure 6, 7: I could not find why the authors show spectrum 1 and 2 in Figure 7. As far as I understand, both spectra were obtained at different point but the same light gray tissue in Figure 6, and they should show same spectrum. Does the comparison of spectrum 1 and 2 mean discussion of homogeneity in segregation region at the same depth?

Spelling inconsistency: effective distribution coefficient KE (formula 2, line 213, 221, 289)

Author Response

Point 1: Line 125-126: In Figure 3 and 4, it seems that the grain size has relatively significant difference between the samples at 150 mm and at 200 mm. I suppose that the micromorphological feature at different depth should be discussed based on Figure 3 and 4, but not Figure 2. Even if the grain size does not linearly correlate with the ultrasonic pressure, it seems that the formula 1 is not enough to interpret the results shown in Figure 3, 4 (e.g. The grain size at different depth under continuous UST, especially at 1500 W, seems to increase at an accelerating rate.).

Response 1: Thank you for your reply and suggestions and I have finished the modification. As you said, we also found that Figure 1 was insufficient to show the change of grain size at the depth of 150 mm and 200 mm. Although grain sizes have change obviously under the power of 1500 W in Figure 3,4. But for our experiment , we considered 1200 W to be a power value in the middle range so we chose the images of microstrcture and segregation under the vibration power of 1200 W as an example.

Point 2: Figure 6, 7: I could not find why the authors show spectrum 1 and 2 in Figure 7. As far as I understand, both spectra were obtained at different point but the same light gray tissue in Figure 6, and they should show same spectrum. Does the comparison of spectrum 1 and 2 mean discussion of homogeneity in segregation region at the same depth?

Response 2: Solute redistribution would inevitably occur during solidification. We also found that the distribution of elements was not completely uniform in segregation. Therefore, we took two points at the same light gray tissues and used their mean values to characterize the element content in segregation to ensure our experiment results are relatively correct.

Point 3: Spelling inconsistency: effective distribution coefficient KE (formula 2, line 213, 221, 289)

Response 3:Thank you for your reply and suggestions and I have finished the modification.

Reviewer 3 Report

On a first sight Figure 2 is a little bit confusing. After understanding the idea behind it (relation between immersion depth and microstructre) and confronting it with Fig. 3 it begs the question why don’t you compare microstructures at the same immersion depth but with different ultrasonic powers. Effect of power input is much greater than effect of immersion depth. But this is only suggestion you might take into consideration if you have proper microstructure pictures.

Author Response

Point 1: On a first sight Figure 2 is a little bit confusing. After understanding the idea behind it (relation between immersion depth and microstructre) and confronting it with Fig. 3 it begs the question why don’t you compare microstructures at the same immersion depth but with different ultrasonic powers. Effect of power input is much greater than effect of immersion depth. But this is only suggestion you might take into consideration if you have proper microstructure pictures.

Response 2: Thank you for your reply and suggestions . As you said, we also found that the influence of ultrasonic power on the micro-morphology was also very significant. There are a lot of researches on ultrasonic power at abroad, but there are relatively few researches on the scope of ultrasonic action. In the actual production, the dimensions of castings are varied. There are inevitably some areas that cannot be affected by ultrasonic waves after ultrasonic vibration was applied.We designed this experiment to explore the scope of ultrasonic action.